# Deep Temporal Sigmoid Belief Networks for Sequence Modeling

**Zhe Gan, Chunyuan Li, Ricardo Henao, David Carlson and Lawrence Carin**
Department of Electrical and Computer Engineering
Duke University, Durham, NC 27708
{zhe.gan, chunyuan.li, r.henao, david.carlson, lcarin}@duke.edu

## Abstract

Deep dynamic generative models are developed to learn sequential dependencies in time-series data. The multi-layered model is designed by constructing a hierarchy of temporal sigmoid belief networks (TSBNs), defined as a sequential stack of sigmoid belief networks (SBNs). Each SBN has a contextual hidden state, inherited from the previous SBNs in the sequence, and is used to regulate its hidden bias. Scalable learning and inference algorithms are derived by introducing a recognition model that yields fast sampling from the variational posterior. This recognition model is trained jointly with the generative model, by maximizing its variational lower bound on the log-likelihood. Experimental results on bouncing balls, polyphonic music, motion capture, and text streams show that the proposed approach achieves state-of-the-art predictive performance, and has the capacity to synthesize various sequences.

## 1 Introduction

Considerable research has been devoted to developing probabilistic models for high-dimensional time-series data, such as video and music sequences, motion capture data, and text streams. Among them, Hidden Markov Models (HMMs) [1] and Linear Dynamical Systems (LDS) [2] have been widely studied, but they may be limited in the type of dynamical structures they can model. An HMM is a mixture model, which relies on a single multinomial variable to represent the history of a time-series. To represent $N$ bits of information about the history, an HMM could require $2^N$ distinct states. On the other hand, real-world sequential data often contain complex non-linear temporal dependencies, while a LDS can only model simple linear dynamics.

Another class of time-series models, which are potentially better suited to model complex probability distributions over high-dimensional sequences, relies on the use of Recurrent Neural Networks (RNNs) [3, 4, 5, 6], and variants of a well-known *undirected* graphical model called the Restricted Boltzmann Machine (RBM) [7, 8, 9, 10, 11]. One such variant is the Temporal Restricted Boltzmann Machine (TRBM) [8], which consists of a sequence of RBMs, where the state of one or more previous RBMs determine the biases of the RBM in the current time step. Learning and inference in the TRBM is non-trivial. The approximate procedure used in [8] is heuristic and not derived from a principled statistical formalism.

Recently, deep *directed* generative models [12, 13, 14, 15] are becoming popular. A *directed* graphical model that is closely related to the RBM is the Sigmoid Belief Network (SBN) [16]. In the work presented here, we introduce the Temporal Sigmoid Belief Network (TSBN), which can be viewed as a temporal stack of SBNs, where each SBN has a contextual hidden state that is inherited from the previous SBNs and is used to adjust its hidden-units bias. Based on this, we further develop a deep dynamic generative model by constructing a hierarchy of TSBNs. This can be considered

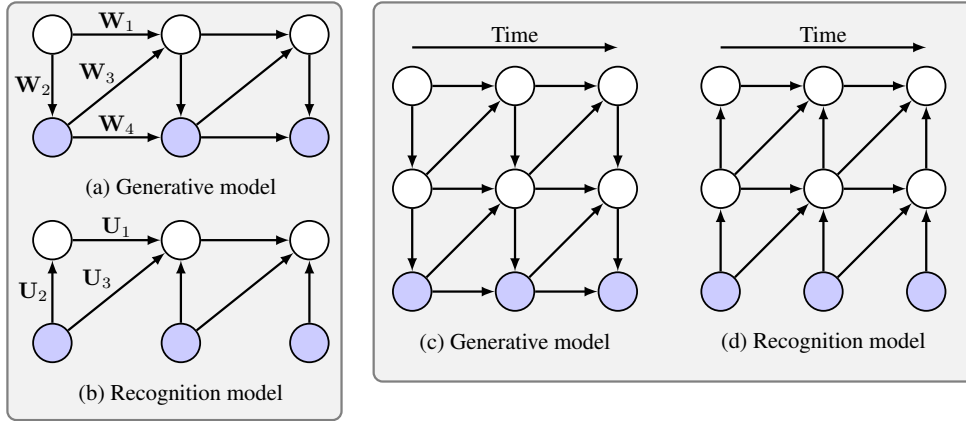

Figure 1: Graphical model for the Deep Temporal Sigmoid Belief Network. (a,b) Generative and recognition model of the TSBN. (c,d) Generative and recognition model of a two-layer Deep TSBN.

as a deep SBN [15] with temporal feedback loops on each layer. Both stochastic and deterministic hidden layers are considered.

Compared with previous work, our model: (*i*) can be viewed as a generalization of an HMM with distributed hidden state representations, and with a deep architecture; (*ii*) can be seen as a generalization of a LDS with complex non-linear dynamics; (*iii*) can be considered as a probabilistic construction of the traditionally deterministic RNN; (*iv*) is closely related to the TRBM, but it has a fully generative process, where data are readily generated from the model using ancestral sampling; (*v*) can be utilized to model different kinds of data, *e.g.*, binary, real-valued and counts.

The "explaining away" effect described in [17] makes inference slow, if one uses traditional inference methods. Another important contribution we present here is to develop fast and scalable learning and inference algorithms, by introducing a recognition model [12, 13, 14], that learns an inverse mapping from observations to hidden variables, based on a loss function derived from a variational principle. By utilizing the recognition model and variance-reduction techniques from [13], we achieve fast inference both at training and testing time.

## 2 Model Formulation

### 2.1 Sigmoid Belief Networks

Deep dynamic generative models are considered, based on the Sigmoid Belief Network (SBN) [16]. An SBN is a Bayesian network that models a binary visible vector $\boldsymbol{v} \in \{0,1\}^M$, in terms of binary hidden variables $\boldsymbol{h} \in \{0,1\}^J$ and weights $\mathbf{W} \in \mathbb{R}^{M \times J}$ with

$$p(v_m = 1|\boldsymbol{h}) = \sigma(\boldsymbol{w}_m^\top \boldsymbol{h} + c_m), \qquad p(h_j = 1) = \sigma(b_j), \tag{1}$$

where $\boldsymbol{v} = [v_1, \ldots, v_M]^\top$, $\boldsymbol{h} = [h_1, \ldots, h_J]^\top$, $\mathbf{W} = [\boldsymbol{w}_1, \ldots, \boldsymbol{w}_M]^\top$, $\boldsymbol{c} = [c_1, \ldots, c_M]^\top$, $\boldsymbol{b} = [b_1, \ldots, b_J]^\top$, and the logistic function, $\sigma(x) \triangleq 1/(1 + e^{-x})$. The parameters $\mathbf{W}$, $\boldsymbol{b}$ and $\boldsymbol{c}$ characterize all data, and the hidden variables, $\boldsymbol{h}$, are specific to particular visible data, $\boldsymbol{v}$.

The SBN is closely related to the RBM [18], which is a Markov random field with the same bipartite structure as the SBN. The RBM defines a distribution over a binary vector that is proportional to the exponential of its *energy*, defined as $-E(\boldsymbol{v}, \boldsymbol{h}) = \boldsymbol{v}^\top \boldsymbol{c} + \boldsymbol{v}^\top \mathbf{W} \boldsymbol{h} + \boldsymbol{h}^\top \boldsymbol{b}$. The conditional distributions, $p(\boldsymbol{v}|\boldsymbol{h})$ and $p(\boldsymbol{h}|\boldsymbol{v})$, in the RBM are factorial, which makes inference fast, while parameter estimation usually relies on an approximation technique known as Contrastive Divergence (CD) [18].

The energy function of an SBN may be written as $-E(\boldsymbol{v}, \boldsymbol{h}) = \boldsymbol{v}^\top \boldsymbol{c} + \boldsymbol{v}^\top \mathbf{W} \boldsymbol{h} + \boldsymbol{h}^\top \boldsymbol{b} - \sum_m \log(1 + \exp(\boldsymbol{w}_m^\top \boldsymbol{h} + c_m))$. SBNs explicitly manifest the generative process to obtain data, in which the hidden layer provides a directed "explanation" for patterns generated in the visible layer. However, the "explaining away" effect described in [17] makes inference inefficient, the latter can be alleviated by exploiting recent advances in variational inference methods [13].

## 2.2 Temporal Sigmoid Belief Networks

The proposed Temporal Sigmoid Belief Network (TSBN) model is a sequence of SBNs arranged in such way that at any given time step, the SBN's biases depend on the state of the SBNs in the previous time steps. Specifically, assume we have a length-$T$ binary visible sequence, the $t$th time step of which is denoted $\boldsymbol{v}_t \in \{0,1\}^M$. The TSBN describes the joint probability as

$$p_{\boldsymbol{\theta}}(\mathbf{V}, \mathbf{H}) = p(\boldsymbol{h}_1)p(\boldsymbol{v}_1|\boldsymbol{h}_1) \cdot \prod_{t=2}^{T} p(\boldsymbol{h}_t|\boldsymbol{h}_{t-1}, \boldsymbol{v}_{t-1}) \cdot p(\boldsymbol{v}_t|\boldsymbol{h}_t, \boldsymbol{v}_{t-1}), \tag{2}$$

where $\mathbf{V} = [\boldsymbol{v}_1, \dots, \boldsymbol{v}_T]$, $\mathbf{H} = [\boldsymbol{h}_1, \dots, \boldsymbol{h}_T]$, and each $\boldsymbol{h}_t \in \{0,1\}^J$ represents the hidden state corresponding to time step $t$. For $t = 1, \dots, T$, each conditional distribution in (2) is expressed as

$$p(h_{jt} = 1|\boldsymbol{h}_{t-1}, \boldsymbol{v}_{t-1}) = \sigma(\boldsymbol{w}_{1j}^\top \boldsymbol{h}_{t-1} + \boldsymbol{w}_{3j}^\top \boldsymbol{v}_{t-1} + b_j), \tag{3}$$

$$p(v_{mt} = 1|\boldsymbol{h}_t, \boldsymbol{v}_{t-1}) = \sigma(\boldsymbol{w}_{2m}^\top \boldsymbol{h}_t + \boldsymbol{w}_{4m}^\top \boldsymbol{v}_{t-1} + c_m), \tag{4}$$

where $\boldsymbol{h}_0$ and $\boldsymbol{v}_0$, needed for the prior model $p(\boldsymbol{h}_1)$ and $p(\boldsymbol{v}_1|\boldsymbol{h}_1)$, are defined as zero vectors, respectively, for conciseness. The model parameters, $\boldsymbol{\theta}$, are specified as $\mathbf{W}_1 \in \mathbb{R}^{J \times J}$, $\mathbf{W}_2 \in \mathbb{R}^{M \times J}$, $\mathbf{W}_3 \in \mathbb{R}^{J \times M}$, $\mathbf{W}_4 \in \mathbb{R}^{M \times M}$. For $i = 1, 2, 3, 4$, $\boldsymbol{w}_{ij}$ is the transpose of the $j$th row of $\mathbf{W}_i$, and $\boldsymbol{c} = [c_1, \dots, c_M]^\top$ and $\boldsymbol{b} = [b_1, \dots, b_J]^\top$ are bias terms. The graphical model for the TSBN is shown in Figure 1(a).

By setting $\mathbf{W}_3$ and $\mathbf{W}_4$ to be zero matrices, the TSBN can be viewed as a Hidden Markov Model [1] with an exponentially large state space, that has a compact parameterization of the transition and the emission probabilities. Specifically, each hidden state in the HMM is represented as a *one-hot* length-$J$ vector, while in the TSBN, the hidden states can be any length-$J$ binary vector. We note that the transition matrix is highly structured, since the number of parameters is only quadratic *w.r.t.* $J$. Compared with the TRBM [8], our TSBN is fully directed, which allows for *fast sampling* of "fantasy" data from the inferred model.

## 2.3 TSBN Variants

**Modeling real-valued data**   The model above can be readily extended to model real-valued sequence data, by substituting (4) with $p(\boldsymbol{v}_t|\boldsymbol{h}_t, \boldsymbol{v}_{t-1}) = \mathcal{N}(\boldsymbol{\mu}_t, \text{diag}(\boldsymbol{\sigma}_t^2))$, where

$$\mu_{mt} = \boldsymbol{w}_{2m}^\top \boldsymbol{h}_t + \boldsymbol{w}_{4m}^\top \boldsymbol{v}_{t-1} + c_m, \quad \log \sigma_{mt}^2 = (\boldsymbol{w}_{2m}')^\top \boldsymbol{h}_t + (\boldsymbol{w}_{4m}')^\top \boldsymbol{v}_{t-1} + c_m', \tag{5}$$

and $\mu_{mt}$ and $\sigma_{mt}^2$ are elements of $\boldsymbol{\mu}_t$ and $\boldsymbol{\sigma}_t^2$, respectively. $\mathbf{W}_2'$ and $\mathbf{W}_4'$ are of the same size of $\mathbf{W}_2$ and $\mathbf{W}_4$, respectively. Compared with the Gaussian TRBM [9], in which $\sigma_{mt}$ is fixed to 1, our formalism uses a diagonal matrix to parameterize the variance structure of $\boldsymbol{v}_t$.

**Modeling count data**   We also introduce an approach for modeling time-series data with count observations, by replacing (4) with $p(\boldsymbol{v}_t|\boldsymbol{h}_t, \boldsymbol{v}_{t-1}) = \prod_{m=1}^{M} y_{mt}^{v_{mt}}$, where

$$y_{mt} = \frac{\exp(\boldsymbol{w}_{2m}^\top \boldsymbol{h}_t + \boldsymbol{w}_{4m}^\top \boldsymbol{v}_{t-1} + c_m)}{\sum_{m'=1}^{M} \exp(\boldsymbol{w}_{2m'}^\top \boldsymbol{h}_t + \boldsymbol{w}_{4m'}^\top \boldsymbol{v}_{t-1} + c_{m'})} . \tag{6}$$

This formulation is related to the Replicated Softmax Model (RSM) described in [19], however, our approach uses a *directed* connection from the binary hidden variables to the visible counts, while also learning the dynamics in the count sequences.

Furthermore, rather than assuming that $\boldsymbol{h}_t$ and $\boldsymbol{v}_t$ only depend on $\boldsymbol{h}_{t-1}$ and $\boldsymbol{v}_{t-1}$, in the experiments, we also allow for connections from the past $n$ time steps of the hidden and visible states, to the current states, $\boldsymbol{h}_t$ and $\boldsymbol{v}_t$. A sliding window is then used to go through the sequence to obtain $n$ frames at each time. We refer to $n$ as the *order* of the model.

## 2.4 Deep Architecture for Sequence Modeling with TSBNs

Learning the sequential dependencies with the shallow model in (2)-(4) may be restrictive. Therefore, we propose two deep architectures to improve its representational power: (*i*) adding stochastic hidden layers; (*ii*) adding deterministic hidden layers. The graphical model for the deep TSBN

is shown in Figure 1(c). Specifically, we consider a deep TSBN with hidden layers $\boldsymbol{h}_t^{(\ell)}$ for $t = 1, \ldots, T$ and $\ell = 1, \ldots, L$. Assume layer $\ell$ contains $J^{(\ell)}$ hidden units, and denote the visible layer $\boldsymbol{v}_t = \boldsymbol{h}_t^{(0)}$ and let $\boldsymbol{h}_t^{(L+1)} = \boldsymbol{0}$, for convenience. In order to obtain a proper generative model, the top hidden layer $\boldsymbol{h}^{(L)}$ contains stochastic binary hidden variables.

For the middle layers, $\ell = 1, \ldots, L-1$, if stochastic hidden layers are utilized, the generative process is expressed as $p(\boldsymbol{h}_t^{(\ell)}) = \prod_{j=1}^{J^{(\ell)}} p(h_{jt}^{(\ell)} | \boldsymbol{h}_t^{(\ell+1)}, \boldsymbol{h}_{t-1}^{(\ell)}, \boldsymbol{h}_{t-1}^{(\ell-1)})$, where each conditional distribution is parameterized via a logistic function, as in (4). If deterministic hidden layers are employed, we obtain $\boldsymbol{h}_t^{(\ell)} = f(\boldsymbol{h}_t^{(\ell+1)}, \boldsymbol{h}_{t-1}^{(\ell)}, \boldsymbol{h}_{t-1}^{(\ell-1)})$, where $f(\cdot)$ is chosen to be a rectified linear function. Although the differences between these two approaches are minor, learning and inference algorithms can be quite different, as shown in Section 3.3.

## 3 Scalable Learning and Inference

Computation of the exact posterior over the hidden variables in (2) is intractable. Approximate Bayesian inference, such as Gibbs sampling or mean-field variational Bayes (VB) inference, can be implemented [15, 16]. However, Gibbs sampling is very inefficient, due to the fact that the conditional posterior distribution of the hidden variables does not factorize. The mean-field VB indeed provides a fully factored variational posterior, but this technique increases the gap between the bound being optimized and the true log-likelihood, potentially resulting in a poor fit to the data. To allow for tractable and scalable inference and parameter learning, without loss of the flexibility of the variational posterior, we apply the *Neural Variational Inference and Learning* (NVIL) algorithm described in [13].

### 3.1 Variational Lower Bound Objective

We are interested in training the TSBN model, $p_{\boldsymbol{\theta}}(\mathbf{V}, \mathbf{H})$, described in (2), with parameters $\boldsymbol{\theta}$. Given an observation $\mathbf{V}$, we introduce a fixed-form distribution, $q_{\boldsymbol{\phi}}(\mathbf{H}|\mathbf{V})$, with parameters $\boldsymbol{\phi}$, that approximates the true posterior distribution, $p(\mathbf{H}|\mathbf{V})$. We then follow the variational principle to derive a lower bound on the marginal log-likelihood, expressed as[1]

$$\mathcal{L}(\mathbf{V}, \boldsymbol{\theta}, \boldsymbol{\phi}) = \mathbb{E}_{q_{\boldsymbol{\phi}}(\mathbf{H}|\mathbf{V})}[\log p_{\boldsymbol{\theta}}(\mathbf{V}, \mathbf{H}) - \log q_{\boldsymbol{\phi}}(\mathbf{H}|\mathbf{V})]. \tag{7}$$

We construct the approximate posterior $q_{\boldsymbol{\phi}}(\mathbf{H}|\mathbf{V})$ as a recognition model. By using this, we avoid the need to compute variational parameters per data point; instead we compute a set of parameters $\boldsymbol{\phi}$ used for all $\mathbf{V}$. In order to achieve fast inference, the recognition model is expressed as

$$q_{\boldsymbol{\phi}}(\mathbf{H}|\mathbf{V}) = q(\boldsymbol{h}_1|\boldsymbol{v}_1) \cdot \prod_{t=2}^{T} q(\boldsymbol{h}_t|\boldsymbol{h}_{t-1}, \boldsymbol{v}_t, \boldsymbol{v}_{t-1}), \tag{8}$$

and each conditional distribution is specified as

$$q(h_{jt} = 1|\boldsymbol{h}_{t-1}, \boldsymbol{v}_t, \boldsymbol{v}_{t-1}) = \sigma(\boldsymbol{u}_{1j}^\top \boldsymbol{h}_{t-1} + \boldsymbol{u}_{2j}^\top \boldsymbol{v}_t + \boldsymbol{u}_{3j}^\top \boldsymbol{v}_{t-1} + d_j), \tag{9}$$

where $\boldsymbol{h}_0$ and $\boldsymbol{v}_0$, for $q(\boldsymbol{h}_1|\boldsymbol{v}_1)$, are defined as zero vectors. The recognition parameters $\boldsymbol{\phi}$ are specified as $\mathbf{U}_1 \in \mathbb{R}^{J \times J}$, $\mathbf{U}_2 \in \mathbb{R}^{J \times M}$, $\mathbf{U}_3 \in \mathbb{R}^{J \times M}$. For $i = 1, 2, 3$, $\boldsymbol{u}_{ij}$ is the transpose of the $j$th row of $\mathbf{U}_i$, and $\boldsymbol{d} = [d_1, \ldots, d_J]^\top$ is the bias term. The graphical model is shown in Figure 1(b).

The recognition model defined in (9) has the same form as in the approximate inference used for the TRBM [8]. Exact inference for our model consists of a forward and backward pass through the entire sequence, that requires the traversing of each possible hidden state. Our feedforward approximation allows the inference procedure to be fast and implemented in an online fashion.

### 3.2 Parameter Learning

To optimize (7), we utilize Monte Carlo methods to approximate expectations and stochastic gradient descent (SGD) for parameter optimization. The gradients can be expressed as

$$\nabla_{\boldsymbol{\theta}} \mathcal{L}(\mathbf{V}) = \mathbb{E}_{q_{\boldsymbol{\phi}}(\mathbf{H}|\mathbf{V})}[\nabla_{\boldsymbol{\theta}} \log p_{\boldsymbol{\theta}}(\mathbf{V}, \mathbf{H})], \tag{10}$$

$$\nabla_{\boldsymbol{\phi}} \mathcal{L}(\mathbf{V}) = \mathbb{E}_{q_{\boldsymbol{\phi}}(\mathbf{H}|\mathbf{V})}[(\log p_{\boldsymbol{\theta}}(\mathbf{V}, \mathbf{H}) - \log q_{\boldsymbol{\phi}}(\mathbf{H}|\mathbf{V})) \times \nabla_{\boldsymbol{\phi}} \log q_{\boldsymbol{\phi}}(\mathbf{H}|\mathbf{V})]. \tag{11}$$

Specifically, in the TSBN model, if we define $\hat{v}_{mt} = \sigma(\boldsymbol{w}_{2m}^\top \boldsymbol{h}_t + \boldsymbol{w}_{4m}^\top \boldsymbol{v}_{t-1} + c_m)$ and $\hat{h}_{jt} = \sigma(\boldsymbol{u}_{1j}^\top \boldsymbol{h}_{t-1} + \boldsymbol{u}_{2j}^\top \boldsymbol{v}_t + \boldsymbol{u}_{3j}^\top \boldsymbol{v}_{t-1} + d_j)$, the gradients for $\boldsymbol{w}_{2m}$ and $\boldsymbol{u}_{2j}$ can be calculated as

$$\frac{\partial \log p_{\boldsymbol{\theta}}(\mathbf{V}, \mathbf{H})}{\partial w_{2mj}} = \sum_{t=1}^{T} (v_{mt} - \hat{v}_{mt}) \cdot h_{jt}, \qquad \frac{\partial \log q_{\boldsymbol{\phi}}(\mathbf{H}|\mathbf{V})}{\partial u_{2jm}} = \sum_{t=1}^{T} (h_{jt} - \hat{h}_{jt}) \cdot v_{mt}. \quad (12)$$

Other update equations, along with the learning details for the TSBN variants in Section 2.3, are provided in the Supplementary Section B. We observe that the gradients in (10) and (11) share many similarities with the *wake-sleep* algorithm [20]. Wake-sleep alternates between updating $\boldsymbol{\theta}$ in the wake phase and updating $\boldsymbol{\phi}$ in the sleep phase. The update of $\boldsymbol{\theta}$ is based on the samples generated from $q_{\boldsymbol{\phi}}(\mathbf{H}|\mathbf{V})$, and is identical to (10). However, in contrast to (11), the recognition parameters $\boldsymbol{\phi}$ are estimated from samples generated by the model, *i.e.*, $\nabla_{\boldsymbol{\phi}} \mathcal{L}(\mathbf{V}) = \mathbb{E}_{p_{\boldsymbol{\theta}}(\mathbf{V},\mathbf{H})}[\nabla_{\boldsymbol{\phi}} \log q_{\boldsymbol{\phi}}(\mathbf{H}|\mathbf{V})]$. This update does not optimize the same objective as in (10), hence the wake-sleep algorithm is not guaranteed to converge [13].

Inspecting (11), we see that we are using $l_{\boldsymbol{\phi}}(\mathbf{V}, \mathbf{H}) = \log p_{\boldsymbol{\theta}}(\mathbf{V}, \mathbf{H}) - \log q_{\boldsymbol{\phi}}(\mathbf{H}|\mathbf{V})$ as the *learning signal* for the recognition parameters $\boldsymbol{\phi}$. The expectation of this learning signal is exactly the lower bound (7), which is easy to evaluate. However, this tractability makes the estimated gradients of the recognition parameters very noisy. In order to make the algorithm practical, we employ the variance reduction techniques proposed in [13], namely: (*i*) centering the learning signal, by subtracting the data-independent baseline and the data-dependent baseline; (*ii*) variance normalization, by dividing the centered learning signal by a running estimate of its standard deviation. The data-dependent baseline is implemented using a neural network. Additionally, RMSprop [21], a form of SGD where the gradients are adaptively rescaled by a running average of their recent magnitude, were found in practice to be important for fast convergence; thus utilized throughout all the experiments. The outline of the NVIL algorithm is provided in the Supplementary Section A.

### 3.3 Extension to deep models

The recognition model corresponding to the deep TSBN is shown in Figure 1(d). Two kinds of deep architectures are discussed in Section 2.4. We illustrate the difference of their learning algorithms in two respects: (*i*) the calculation of the lower bound; and (*ii*) the calculation of the gradients.

The top hidden layer is stochastic. If the middle hidden layers are also stochastic, the calculation of the lower bound is more involved, compared with the shallow model; however, the gradient evaluation remain simple as in (12). On the other hand, if deterministic middle hidden layers (*i.e.*, recurrent neural networks) are employed, the lower bound objective will stay the same as a shallow model, since the only stochasticity in the generative process lies in the top layer; however, the gradients have to be calculated recursively through the *back-propagation through time* algorithm [22]. All details are provided in the Supplementary Section C.

## 4 Related Work

The RBM has been widely used as building block to learn the sequential dependencies in time-series data, *e.g.*, the conditional-RBM-related models [7, 23], and the temporal RBM [8]. To make *exact* inference possible, the recurrent temporal RBM was also proposed [9], and further extended to learn the dependency structure within observations [11].

In the work reported here, we focus on modeling sequences based on the SBN [16], which recently has been shown to have the potential to build deep generative models [13, 15, 24]. Our work serves as another extension of the SBN that can be utilized to model time-series data. Similar ideas have also been considered in [25] and [26]. However, in [25], the authors focus on grammar learning, and use a feed-forward approximation of the mean-field VB to carry out the inference; while in [26], the wake-sleep algorithm was developed. We apply the model in a different scenario, and develop a fast and scalable inference algorithm, based on the idea of training a recognition model by leveraging the stochastic gradient of the variational bound.

There exist two main methods for the training of recognition models. The first one, termed Stochastic Gradient Variational Bayes (SGVB), is based on a reparameterization trick [12, 14], which can be only employed in models with continuous latent variables, *e.g.*, the variational auto-encoder [12]

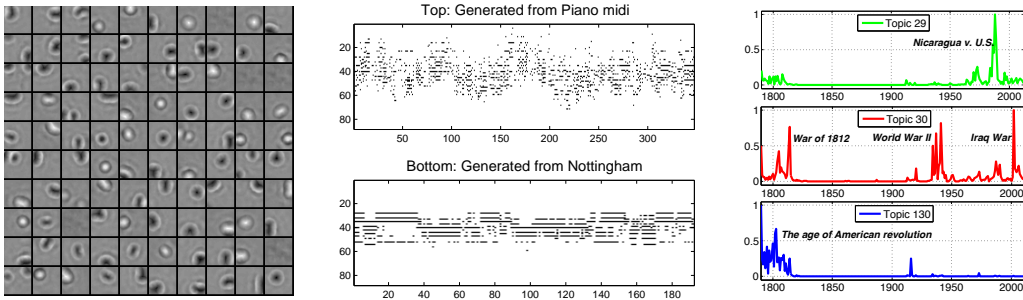

Figure 2: (Left) Dictionaries learned using the HMSBN for the videos of bouncing balls. (Middle) Samples generated from the HMSBN trained on the polyphonic music. Each column is a sample vector of notes. (Right) Time evolving from 1790 to 2014 for three selected topics learned from the STU dataset. Plotted values represent normalized probabilities that the topic appears in a given year. Best viewed electronically.

and all the recent recurrent extensions of it [27, 28, 29]. The second one, called Neural Variational Inference and Learning (NVIL), is based on the log-derivative trick [13], which is more general and can also be applicable to models with discrete random variables. The NVIL algorithm has been previously applied to the training of SBN in [13]. Our approach serves as a new application of this algorithm for a SBN-based time-series model.

## 5   Experiments

We present experimental results on four publicly available datasets: the bouncing balls [9], polyphonic music [10], motion capture [7] and state-of-the-Union [30]. To assess the performance of the TSBN model, we show sequences generated from the model, and report the average log-probability that the model assigns to a test sequence, and the average squared one-step-ahead prediction error per frame. Code is available at `https://github.com/zhegan27/TSBN_code_NIPS2015`.

The TSBN model with $\mathbf{W}_3 = \mathbf{0}$ and $\mathbf{W}_4 = \mathbf{0}$ is denoted Hidden Markov SBN (HMSBN), the deep TSBN with stochastic hidden layer is denoted DTSBN-S, and the deep TSBN with deterministic hidden layer is denoted DTSBN-D.

Model parameters were initialized by sampling randomly from $\mathcal{N}(\mathbf{0}, 0.001^2 \mathbf{I})$, except for the bias parameters, that were initialized as 0. The TSBN model is trained using a variant of RMSprop [6], with momentum of 0.9, and a constant learning rate of $10^{-4}$. The decay over the root mean squared gradients is set to 0.95. The maximum number of iterations we use is $10^5$. The gradient estimates were computed using a single sample from the recognition model. The only regularization we used was a weight decay of $10^{-4}$. The data-dependent baseline was implemented by using a neural network with a single hidden layer with 100 tanh units.

For the prediction of $\boldsymbol{v}_t$ given $\boldsymbol{v}_{1:t-1}$, we (*i*) first obtain a sample from $q_{\boldsymbol{\phi}}(\boldsymbol{h}_{1:t-1}|\boldsymbol{v}_{1:t-1})$; (*ii*) calculate the conditional posterior $p_{\boldsymbol{\theta}}(\boldsymbol{h}_t|\boldsymbol{h}_{1:t-1}, \boldsymbol{v}_{1:t-1})$ of the current hidden state ; (*iii*) make a prediction for $\boldsymbol{v}_t$ using $p_{\boldsymbol{\theta}}(\boldsymbol{v}_t|\boldsymbol{h}_{1:t}, \boldsymbol{v}_{1:t-1})$. On the other hand, synthesizing samples is conceptually simper. Sequences can be readily generated from the model using ancestral sampling.

### 5.1   Bouncing balls dataset

We conducted the first experiment on synthetic videos of 3 bouncing balls, where pixels are binary valued. We followed the procedure in [9], and generated 4000 videos for training, and another 200 videos for testing. Each video is of length 100 and of resolution $30 \times 30$.

The dictionaries learned using the HMSBN are shown in Figure 2 (Left). Compared with previous work [9, 10], our learned bases are more spatially localized. In Table 1, we compare the average squared prediction error per frame over the 200 test videos, with recurrent temporal RBM (RTRBM) and structured RTRBM (SRTRBM). As can be seen, our approach achieves better performance compared with the baselines in the literature. Furthermore, we observe that a high-order TSBN reduces the prediction error significantly, compared with an order-one TSBN. This is due to the fact

Table 1: Average prediction error for the bouncing balls dataset. (◇) taken from [11].

| MODEL | DIM | ORDER | PRED. ERR. |
|---|---|---|---|
| DTSBN-S | 100-100 | 2 | **2.79** ± 0.39 |
| DTSBN-D | 100-100 | 2 | 2.99 ± 0.42 |
| TSBN | 100 | 4 | 3.07 ± 0.40 |
| TSBN | 100 | 1 | 9.48 ± 0.38 |
| RTRBM◇ | 3750 | 1 | 3.88 ± 0.33 |
| SRTRBM◇ | 3750 | 1 | 3.31 ± 0.33 |

Table 2: Average prediction error obtained for the motion capture dataset. (◇) taken from [11].

| MODEL | WALKING | RUNNING |
|---|---|---|
| DTSBN-S | **4.40** ± 0.28 | **2.56** ± 0.40 |
| DTSBN-D | 4.62 ± 0.01 | 2.84 ± 0.01 |
| TSBN | 5.12 ± 0.50 | 4.85 ± 1.26 |
| HMSBN | 10.77 ± 1.15 | 7.39 ± 0.47 |
| SS-SRTRBM◇ | 8.13 ± 0.06 | 5.88 ± 0.05 |
| G-RTRBM◇ | 14.41 ± 0.38 | 10.91 ± 0.27 |

that by using a high-order TSBN, more information about the past is conveyed. We also examine the advantage of employing deep models. Using stochastic, or deterministic hidden layer improves performances. More results, including log-likelihoods, are provided in Supplementary Section D.

## 5.2 Motion capture dataset

In this experiment, we used the CMU motion capture dataset, that consists of measured joint angles for different motion types. We used the 33 running and walking sequences of subject 35 (23 walking sequences and 10 running sequences). We followed the preprocessing procedure of [11], after which we were left with 58 joint angles. We partitioned the 33 sequences into training and testing set: the first of which had 31 sequences, and the second had 2 sequences (one walking and another running). We averaged the prediction error over 100 trials, as reported in Table 2. The TSBN we implemented is of size 100 in each hidden layer and order 1. It can be seen that the TSBN-based models improves over the Gaussian (G-)RTRBM and the spike-slab (SS-)SRTRBM significantly.

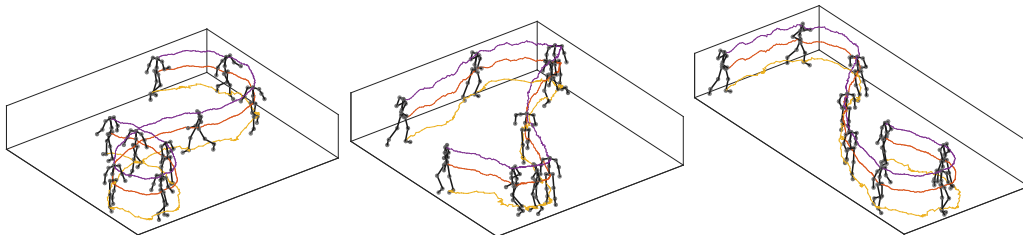

Figure 3: Motion trajectories generated from the HMSBN trained on the motion capture dataset. (Left) Walking. (Middle) Running-running-walking. (Right) Running-walking.

Another popular motion capture dataset is the MIT dataset[2]. To further demonstrate the directed, generative nature of our model, we give our trained HMSBN model different initializations, and show generated, synthetic data and the transitions between different motion styles in Figure 3. These generated data are readily produced from the model and demonstrate realistic behavior. The smooth trajectories are walking movements, while the vibrating ones are running. Corresponding video files (AVI) are provided as mocap 1, 2 and 3 in the Supplementary Material.

## 5.3 Polyphonic music dataset

The third experiment is based on four different polyphonic music sequences of piano [10], *i.e.*, Piano-midi.de (Piano), Nottingham (Nott), MuseData (Muse) and JSB chorales (JSB). Each of these datasets are represented as a collection of 88-dimensional binary sequences, that span the whole range of piano from A0 to C8.

The samples generated from the trained HMSBN model are shown in Figure 2 (Middle). As can be seen, different styles of polyphonic music are synthesized. The corresponding MIDI files are provided as music 1 and 2 in the Supplementary Material. Our model has the ability to learn basic harmony rules and local temporal coherence. However, long-term structure and musical melody remain elusive. The variational lower bound, along with the estimated log-likelihood in [10], are presented in Table 3. The TSBN we implemented is of size 100 and order 1. Empirically, adding layers did not improve performance on this dataset, hence no such results are reported. The results of RNN-NADE and RTRBM [10] were obtained by only 100 runs of the annealed importance sampling, which has the potential to overestimate the true log-likelihood. Our variational lower bound provides a more conservative estimate. Though, our performance is still better than that of RNN.

Table 3: Test log-likelihood for the polyphonic music dataset. (⋄) taken from [10].

| MODEL | PIANO. | NOTT. | MUSE. | JSB. |
|---|---|---|---|---|
| TSBN | -7.98 | -3.67 | -6.81 | -7.48 |
| RNN-NADE⋄ | **-7.05** | **-2.31** | **-5.60** | **-5.56** |
| RTRBM⋄ | -7.36 | -2.62 | -6.35 | -6.35 |
| RNN⋄ | -8.37 | -4.46 | -8.13 | -8.71 |

Table 4: Average prediction precision for STU. (⋄) taken from [31].

| MODEL | DIM | MP | PP |
|---|---|---|---|
| HMSBN | 25 | **0.327** ± 0.002 | 0.353 ± 0.070 |
| DHMSBN-s | 25-25 | 0.299 ± 0.001 | **0.378** ± 0.006 |
| GP-DPFA ⋄ | 100 | 0.223 ± 0.001 | 0.189 ± 0.003 |
| DRFM⋄ | 25 | 0.217 ± 0.003 | 0.177 ± 0.010 |

## 5.4 State of the Union dataset

The State of the Union (STU) dataset contains the transcripts of $T = 225$ US State of the Union addresses, from 1790 to 2014. Two tasks are considered, *i.e.*, prediction and dynamic topic modeling.

**Prediction** The prediction task is concerned with estimating the held-out words. We employ the setup in [31]. After removing stop words and terms that occur fewer than 7 times in one document or less than 20 times overall, there are 2375 unique words. The entire data of the last year is held-out. For the documents in the previous years, we randomly partition the words of each document into 80%/20% split. The model is trained on the 80% portion, and the remaining 20% held-out words are used to test the prediction at each year. The words in both held-out sets are ranked according to the probability estimated from (6).

To evaluate the prediction performance, we calculate the precision @top-$M$ as in [31], which is given by the fraction of the top-$M$ words, predicted by the model, that matches the true ranking of the word counts. $M = 50$ is used. Two recent works are compared, GP-DPFA [31] and DRFM [30]. The results are summarized in Table 4. Our model is of order 1. The column MP denotes the mean precision over all the years that appear in the training set. The column PP denotes the predictive precision for the final year. Our model achieves significant improvements in both scenarios.

**Dynamic Topic Modeling** The setup described in [30] is employed, and the number of topics is 200. To understand the temporal dynamic per topic, three topics are selected and the normalized probability that a topic appears at each year are shown in Figure 2 (Right). Their associated top 6 words per topic are shown in Table 5. The learned trajectory exhibits different temporal patterns across the topics. Clearly, we can identify jumps associated with some key historical events. For instance, for Topic 29, we observe a positive jump in 1986 related to military and paramilitary activities in and against Nicaragua brought by the U.S. Topic 30 is related with war, where the War of 1812, World War II and Iraq War all spike up in their corresponding years. In Topic 130, we observe consistent positive jumps from 1890 to 1920, when the American revolution was taking place. Three other interesting topics are also shown in Table 5. Topic 64 appears to be related to education, Topic 70 is about Iraq, and Topic 74 is Axis and World War II. We note that the words for these topics are explicitly related to these matters.

Table 5: Top 6 most probable words associated with the STU topics.

| Topic #29 | Topic #30 | Topic #130 | Topic #64 | Topic #70 | Topic #74 |
|---|---|---|---|---|---|
| family | officer | government | generations | Iraqi | Philippines |
| budget | civilized | country | generation | Qaida | islands |
| Nicaragua | warfare | public | recognize | Iraq | axis |
| free | enemy | law | brave | Iraqis | Nazis |
| future | whilst | present | crime | AI | Japanese |
| freedom | gained | citizens | race | Saddam | Germans |

## 6 Conclusion

We have presented the Deep Temporal Sigmoid Belief Networks, an extension of SBN, that models the temporal dependencies in high-dimensional sequences. To allow for scalable inference and learning, an efficient variational optimization algorithm is developed. Experimental results on several datasets show that the proposed approach obtains superior predictive performance, and synthesizes interesting sequences.

In this work, we have investigated the modeling of different types of data individually. One interesting future work is to combine them into a unified framework for dynamic multi-modality learning. Furthermore, we can use high-order optimization methods to speed up inference [32].

**Acknowledgements** This research was supported in part by ARO, DARPA, DOE, NGA and ONR.

## Footnotes

[1]This lower bound is equivalent to the marginal log-likelihood if $q_{\boldsymbol{\phi}}(\mathbf{H}|\mathbf{V}) = p(\mathbf{H}|\mathbf{V})$.

[2]Quantitative results on the MIT dataset are provided in Supplementary Section D.

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
