[Supplementary Material · TSBNsupp_NIPS2015.pdf]

# Deep Temporal Sigmoid Belief Networks for Sequence Modeling: Supplementary Material

**Zhe Gan, Chunyuan Li, Ricardo Henao, David Carlson and Lawrence Carin**
Department of Electrical and Computer Engineering
Duke University, Durham, NC 27708
{zhe.gan, chunyuan.li, r.henao, david.carlson, lcarin}@duke.edu

## A   Outline of the NVIL algorithm

The outline of the NVIL algorithm for computing gradients are shown below (reproduced from [1]). $C_{\boldsymbol{\lambda}}(\boldsymbol{v}_t)$ represents the data-dependent baseline, and $\alpha = 0.8$ throughout the experiments.

---

**Algorithm 1** Compute gradient estimates for the model parameters and recognition parameters.

---

$\Delta\boldsymbol{\theta} \leftarrow 0, \Delta\boldsymbol{\phi} \leftarrow 0, \Delta\boldsymbol{\lambda} \leftarrow 0$
$\mathcal{L} \leftarrow 0$
**for** $t \leftarrow 1$ **to** $T$ **do**
  $\boldsymbol{h}_t \sim q_{\boldsymbol{\phi}}(\boldsymbol{h}_t|\boldsymbol{v}_t)$
  $l_t \leftarrow \log p_{\boldsymbol{\theta}}(\boldsymbol{v}_t, \boldsymbol{h}_t) - \log q_{\boldsymbol{\phi}}(\boldsymbol{h}_t|\boldsymbol{v}_t)$
  $\mathcal{L} \leftarrow \mathcal{L} + l_t$
  $l_t \leftarrow l_t - C_{\boldsymbol{\lambda}}(\boldsymbol{v}_t)$
**end for**
$c_b \leftarrow \text{mean}(l_1, \ldots, l_T)$
$v_b \leftarrow \text{variance}(l_1, \ldots, l_T)$
$c \leftarrow \alpha c + (1 - \alpha)c_b$
$v \leftarrow \alpha v + (1 - \alpha)v_b$
**for** $t \leftarrow 1$ **to** $T$ **do**
  $l_t \leftarrow \frac{l_t - c}{\max(1, \sqrt{v})}$
  $\Delta\boldsymbol{\theta} \leftarrow \Delta\boldsymbol{\theta} + \nabla_{\boldsymbol{\theta}} \log p_{\boldsymbol{\theta}}(\boldsymbol{v}_t, \boldsymbol{h}_t)$
  $\Delta\boldsymbol{\phi} \leftarrow \Delta\boldsymbol{\phi} + l_t \nabla_{\boldsymbol{\phi}} \log q_{\boldsymbol{\phi}}(\boldsymbol{h}_t|\boldsymbol{v}_t)$
  $\Delta\boldsymbol{\lambda} \leftarrow \Delta\boldsymbol{\lambda} + l_t \nabla_{\boldsymbol{\lambda}} C_{\boldsymbol{\lambda}}(\boldsymbol{v}_t)$
**end for**

---

## B   Learning and Inference Details on TSBN

For $t = 1, \ldots, T$, consider $\boldsymbol{v}_t \in \{0,1\}^M$, $\boldsymbol{h}_t \in \{0,1\}^J$, the model parameters $\boldsymbol{\theta}$ are specified as $\mathbf{W}_1 \in \mathbb{R}^{J \times J}$, $\mathbf{W}_2 \in \mathbb{R}^{M \times J}$, $\mathbf{W}_3 \in \mathbb{R}^{J \times M}$, $\mathbf{W}_4 \in \mathbb{R}^{M \times M}$, $\boldsymbol{b} \in \mathbb{R}^J$, and $\boldsymbol{c} \in \mathbb{R}^M$. The generative model is expressed as

$$p(h_{jt} = 1|\boldsymbol{h}_{t-1}, \boldsymbol{v}_{t-1}) = \sigma(\boldsymbol{w}_{1j}^\top \boldsymbol{h}_{t-1} + \boldsymbol{w}_{3j}^\top \boldsymbol{v}_{t-1} + b_j), \tag{1}$$

$$p(v_{mt} = 1|\boldsymbol{h}_t, \boldsymbol{v}_{t-1}) = \sigma(\boldsymbol{w}_{2m}^\top \boldsymbol{h}_t + \boldsymbol{w}_{4m}^\top \boldsymbol{v}_{t-1} + c_m), \tag{2}$$

The recognition model is expressed as

$$q(h_{jt} = 1|\boldsymbol{h}_{t-1}, \boldsymbol{v}_t, \boldsymbol{v}_{t-1}) = \sigma(\boldsymbol{u}_{1j}^\top \boldsymbol{h}_{t-1} + \boldsymbol{u}_{2j}^\top \boldsymbol{v}_t + \boldsymbol{u}_{3j}^\top \boldsymbol{v}_{t-1} + d_j), \tag{3}$$

where the recognition parameters are specified as $\mathbf{U}_1 \in \mathbb{R}^{J \times J}$, $\mathbf{U}_2 \in \mathbb{R}^{J \times M}$, $\mathbf{U}_3 \in \mathbb{R}^{J \times M}$, and $\boldsymbol{d} \in \mathbb{R}^J$. $\boldsymbol{h}_0$ and $\boldsymbol{v}_0$, needed for $p(\boldsymbol{h}_1)$, $p(\boldsymbol{v}_1|\boldsymbol{h}_1)$ and $q(\boldsymbol{h}_1|\boldsymbol{v}_1)$, are defined as zero vectors, for conciseness.

In order to implement the NVIL algorithm described in [1], we need to calculate the lower bound and also the gradients. Specifically, we have the variational lower bound $\mathcal{L} = \sum_{t=1}^{T} \mathbb{E}_{q_\phi(\boldsymbol{h}|\boldsymbol{v})}[l_t]$, where $l_t$ is expressed as

$$l_t = \sum_{j=1}^{J} \left( \psi_{jt}^{(1)} h_{jt} - \log(1 + \exp(\psi_{jt}^{(1)})) \right) + \sum_{m=1}^{M} \left( \psi_{mt}^{(2)} v_{mt} - \log(1 + \exp(\psi_{mt}^{(2)})) \right) \quad (4)$$

$$- \left[ \sum_{j=1}^{J} \left( \psi_{jt}^{(3)} h_{jt} - \log(1 + \exp(\psi_{jt}^{(3)})) \right) \right] ,$$

and we have defined

$$\psi_{jt}^{(1)} = \boldsymbol{w}_{1j}^\top \boldsymbol{h}_{t-1} + \boldsymbol{w}_{3j}^\top \boldsymbol{v}_{t-1} + b_j , \quad (5)$$

$$\psi_{mt}^{(2)} = \boldsymbol{w}_{2m}^\top \boldsymbol{h}_t + \boldsymbol{w}_{4m}^\top \boldsymbol{v}_{t-1} + c_m , \quad (6)$$

$$\psi_{jt}^{(3)} = \boldsymbol{u}_{1j}^\top \boldsymbol{h}_{t-1} + \boldsymbol{u}_{2j}^\top \boldsymbol{v}_t + \boldsymbol{u}_{3j}^\top \boldsymbol{v}_{t-1} + d_j . \quad (7)$$

By further defining

$$\chi_{jt}^{(1)} = h_{jt} - \sigma(\psi_{jt}^{(1)}), \quad \chi_{mt}^{(2)} = v_{mt} - \sigma(\psi_{mt}^{(2)}), \quad \chi_{jt}^{(3)} = h_{jt} - \sigma(\psi_{jt}^{(3)}), \quad (8)$$

The gradients for the model parameters $\boldsymbol{\theta}$ are expressed as

$$\frac{\partial \log p_{\boldsymbol{\theta}}(\boldsymbol{v}_t, \boldsymbol{h}_t)}{\partial w_{1jj'}} = \chi_{jt}^{(1)} h_{j't-1}, \quad \frac{\partial \log p_{\boldsymbol{\theta}}(\boldsymbol{v}_t, \boldsymbol{h}_t)}{\partial w_{3jm}} = \chi_{jt}^{(1)} v_{mt-1}, \quad \frac{\partial \log p_{\boldsymbol{\theta}}(\boldsymbol{v}_t, \boldsymbol{h}_t)}{\partial b_j} = \chi_{jt}^{(1)} , \quad (9)$$

$$\frac{\partial \log p_{\boldsymbol{\theta}}(\boldsymbol{v}_t, \boldsymbol{h}_t)}{\partial w_{2mj}} = \chi_{mt}^{(2)} h_{jt}, \quad \frac{\partial \log p_{\boldsymbol{\theta}}(\boldsymbol{v}_t, \boldsymbol{h}_t)}{\partial w_{4mm'}} = \chi_{mt}^{(2)} v_{m't-1}, \quad \frac{\partial \log p_{\boldsymbol{\theta}}(\boldsymbol{v}_t, \boldsymbol{h}_t)}{\partial c_m} = \chi_{mt}^{(2)} . \quad (10)$$

The gradients for the recognition parameters $\phi$ are expressed as

$$\frac{\partial \log q_{\phi}(\boldsymbol{h}_t|\boldsymbol{v}_t)}{\partial u_{1jj'}} = \chi_{jt}^{(3)} h_{j't-1}, \quad \frac{\partial \log q_{\phi}(\boldsymbol{h}_t|\boldsymbol{v}_t)}{\partial u_{2jm}} = \chi_{jt}^{(3)} v_{mt} , \quad (11)$$

$$\frac{\partial \log q_{\phi}(\boldsymbol{h}_t|\boldsymbol{v}_t)}{\partial u_{3jm}} = \chi_{jt}^{(3)} v_{mt-1}, \quad \frac{\partial \log q_{\phi}(\boldsymbol{h}_t|\boldsymbol{v}_t)}{\partial d_j} = \chi_{jt}^{(3)} . \quad (12)$$

## B.1 Modeling Real-valued Data

When modeling real-valued data, we substitute (2) with $p(\boldsymbol{v}_t|\boldsymbol{h}_t, \boldsymbol{v}_{t-1}) = \mathcal{N}(\boldsymbol{\mu}_t, \mathrm{diag}(\boldsymbol{\sigma}_t^2))$, where

$$\mu_{mt} = \boldsymbol{w}_{2m}^\top \boldsymbol{h}_t + \boldsymbol{w}_{4m}^\top \boldsymbol{v}_{t-1} + c_m, \quad \log \sigma_{mt} = (\boldsymbol{w}_{2m}')^\top \boldsymbol{h}_t + (\boldsymbol{w}_{4m}')^\top \boldsymbol{v}_{t-1} + c_m', \quad (13)$$

and we have $\mathbf{W}_2' \in \mathbb{R}^{M \times J}$ and $\mathbf{W}_4' \in \mathbb{R}^{M \times M}$. The recognition model remains the same as in (3). Let $\tau_{mt} = \log \sigma_{mt}$, we obtain

$$l_t = \sum_{j=1}^{J} \left( \psi_{jt}^{(1)} h_{jt} - \log(1 + \exp(\psi_{jt}^{(1)})) \right) - \sum_{m=1}^{M} \left( \frac{1}{2} \log 2\pi + \tau_{mt} + \frac{(v_{mt} - \mu_{mt})^2}{2e^{2\tau_{mt}}} \right) \quad (14)$$

$$- \left[ \sum_{j=1}^{J} \left( \psi_{jt}^{(3)} h_{jt} - \log(1 + \exp(\psi_{jt}^{(3)})) \right) \right] .$$

All the gradient calculation remains the same as (9)-(12), except the following.

$$\frac{\partial \log p_{\boldsymbol{\theta}}(\boldsymbol{v}_t, \boldsymbol{h}_t)}{\partial w_{2mj}} = \chi_{mt}^{(4)} h_{jt}, \quad \frac{\partial \log p_{\boldsymbol{\theta}}(\boldsymbol{v}_t, \boldsymbol{h}_t)}{\partial w_{4mm'}} = \chi_{mt}^{(4)} v_{m't-1}, \quad \frac{\partial \log p_{\boldsymbol{\theta}}(\boldsymbol{v}_t, \boldsymbol{h}_t)}{\partial c_m} = \chi_{mt}^{(4)} , \quad (15)$$

$$\frac{\partial \log p_{\boldsymbol{\theta}}(\boldsymbol{v}_t, \boldsymbol{h}_t)}{\partial w_{2mj}'} = \chi_{mt}^{(5)} h_{jt}, \quad \frac{\partial \log p_{\boldsymbol{\theta}}(\boldsymbol{v}_t, \boldsymbol{h}_t)}{\partial w_{4mm'}'} = \chi_{mt}^{(5)} v_{m't-1}, \quad \frac{\partial \log p_{\boldsymbol{\theta}}(\boldsymbol{v}_t, \boldsymbol{h}_t)}{\partial c_m'} = \chi_{mt}^{(5)} , \quad (16)$$

where we have defined

$$\chi_{mt}^{(4)} = \frac{\partial \log p_{\boldsymbol{\theta}}(\boldsymbol{v}_t, \boldsymbol{h}_t)}{\partial \mu_{mt}} = \frac{v_{mt} - \mu_{mt}}{e^{2\tau_{mt}}}, \quad \chi_{mt}^{(5)} = \frac{\partial \log p_{\boldsymbol{\theta}}(\boldsymbol{v}_t, \boldsymbol{h}_t)}{\partial \tau_{mt}} = \frac{(v_{mt} - \mu_{mt})^2}{e^{2\tau_{mt}}} - 1 . \quad (17)$$

## B.2 Modeling Count Data

We also introduce an approach for modeling time-series data with count observations, by replacing (2) with $p(\boldsymbol{v}_t|\boldsymbol{h}_t, \boldsymbol{v}_{t-1}) = \prod_{m=1}^{M} y_{mt}^{v_{mt}}$, where

$$y_{mt} = \frac{\exp(\boldsymbol{w}_{2m}^{\top}\boldsymbol{h}_t + \boldsymbol{w}_{4m}^{\top}\boldsymbol{v}_{t-1} + c_m)}{\sum_{m'=1}^{M} \exp(\boldsymbol{w}_{2m'}^{\top}\boldsymbol{h}_t + \boldsymbol{w}_{4m'}^{\top}\boldsymbol{v}_{t-1} + c_{m'})} \ . \tag{18}$$

The recognition model still remains the same as in (3). The $l_t$ now is expressed as

$$l_t = \sum_{j=1}^{J}\left(\psi_{jt}^{(1)}h_{jt} - \log(1 + \exp(\psi_{jt}^{(1)}))\right) + \sum_{m=1}^{M}\left(\psi_{mt}^{(2)}v_{mt} - v_{mt}\log\sum_{m'=1}^{M}\exp(\psi_{mt}^{(2)})\right) \tag{19}$$

$$- \left[\sum_{j=1}^{J}\left(\psi_{jt}^{(3)}h_{jt} - \log(1 + \exp(\psi_{jt}^{(3)}))\right)\right] \ .$$

All the gradient calculations remain the same as (9)-(12), except the following

$$\frac{\partial \log p_{\boldsymbol{\theta}}(\boldsymbol{v}_t, \boldsymbol{h}_t)}{\partial w_{2mj}} = \chi_{mt}^{(6)}h_{jt}, \quad \frac{\partial \log p_{\boldsymbol{\theta}}(\boldsymbol{v}_t, \boldsymbol{h}_t)}{\partial w_{4mm'}} = \chi_{mt}^{(6)}v_{m't-1}, \quad \frac{\partial \log p_{\boldsymbol{\theta}}(\boldsymbol{v}_t, \boldsymbol{h}_t)}{\partial c_m} = \chi_{mt}^{(6)}. \tag{20}$$

where we have defined $\chi_{mt}^{(6)} = v_{mt} - y_{mt}\sum_{m'=1}^{M}v_{m't}$.

# C  Learning and Inference Details on Deep TSBN

For the ease of notation, we consider a two-hidden-layer deep TSBN here, which can be readily extended to a deep model with any depth. For $t = 1, \ldots, T$, we consider the observation as $\boldsymbol{v}_t \in \{0, 1\}^M$. The top hidden layer is denoted as $\boldsymbol{z}_t \in \{0, 1\}^J$.

(a) Generative model          (b) Recognition model

Figure 1: Generative and recognition model of a two-layer Deep TSBN.

## C.1  Using stochastic hidden layer

Denote the first stochastic hidden layer as $\boldsymbol{h}_t \in \{0, 1\}^K$. The generative model is expressed as

$$p(z_{jt} = 1) = \sigma(\boldsymbol{w}_{1j}^{\top}\boldsymbol{z}_{t-1} + \boldsymbol{w}_{3j}^{\top}\boldsymbol{h}_{t-1} + b_{1j}), \tag{21}$$

$$p(h_{kt} = 1) = \sigma(\boldsymbol{w}_{2k}^{\top}\boldsymbol{z}_t + \boldsymbol{w}_{4k}^{\top}\boldsymbol{h}_{t-1} + \boldsymbol{w}_{6k}^{\top}\boldsymbol{v}_{t-1} + b_{2k}), \tag{22}$$

$$p(v_{mt} = 1) = \sigma(\boldsymbol{w}_{5m}^{\top}\boldsymbol{h}_t + \boldsymbol{w}_{7m}^{\top}\boldsymbol{v}_{t-1} + b_{3m}), \tag{23}$$

where we have defined $\mathbf{W}_1 \in \mathbb{R}^{J\times J}$, $\mathbf{W}_2 \in \mathbb{R}^{K\times J}$, $\mathbf{W}_3 \in \mathbb{R}^{J\times K}$, $\mathbf{W}_4 \in \mathbb{R}^{K\times K}$, $\mathbf{W}_5 \in \mathbb{R}^{M\times K}$, $\mathbf{W}_6 \in \mathbb{R}^{K\times M}$, and $\mathbf{W}_7 \in \mathbb{R}^{M\times M}$. The bias terms are $\boldsymbol{b}_1 \in \mathbb{R}^{J\times 1}$, $\boldsymbol{b}_2 \in \mathbb{R}^{K\times 1}$ and $\boldsymbol{b}_3 \in \mathbb{R}^{M\times 1}$. The corresponding recognition model is expressed as

$$q(h_{kt} = 1) = \sigma(\boldsymbol{u}_{5k}^{\top}\boldsymbol{v}_t + \boldsymbol{u}_{4k}^{\top}\boldsymbol{h}_{t-1} + \boldsymbol{u}_{6k}^{\top}\boldsymbol{v}_{t-1} + c_{2k}) \tag{24}$$

$$q(z_{jt} = 1) = \sigma(\boldsymbol{u}_{2j}^{\top}\boldsymbol{h}_t + \boldsymbol{u}_{1j}^{\top}\boldsymbol{z}_{t-1} + \boldsymbol{u}_{3j}^{\top}\boldsymbol{h}_{t-1} + c_{1j}) \tag{25}$$

where the recognition parameters are specified as $\mathbf{U}_1 \in \mathbb{R}^{J \times J}$, $\mathbf{U}_2 \in \mathbb{R}^{J \times K}$, $\mathbf{U}_3 \in \mathbb{R}^{J \times K}$, $\mathbf{U}_4 \in \mathbb{R}^{K \times K}$, $\mathbf{U}_5 \in \mathbb{R}^{K \times M}$ and $\mathbf{U}_6 \in \mathbb{R}^{K \times M}$. The bias terms are $\boldsymbol{c}_1 \in \mathbb{R}^{J \times 1}$ and $\boldsymbol{c}_2 \in \mathbb{R}^{K \times 1}$. Now, $l_t$ is expressed as

$$l_t = \sum_{j=1}^{J} \left( \psi_{jt}^{(1)} z_{jt} - \log(1 + \exp(\psi_{jt}^{(1)})) \right) + \sum_{k=1}^{K} \left( \psi_{kt}^{(2)} h_{kt} - \log(1 + \exp(\psi_{kt}^{(2)})) \right)$$

$$+ \sum_{m=1}^{M} \left( \psi_{mt}^{(3)} v_{mt} - \log(1 + \exp(\psi_{mt}^{(3)})) \right) \tag{26}$$

$$- \left[ \sum_{k=1}^{K} \left( \psi_{kt}^{(4)} h_{kt} - \log(1 + \exp(\psi_{kt}^{(4)})) \right) + \sum_{j=1}^{J} \left( \psi_{jt}^{(5)} z_{jt} - \log(1 + \exp(\psi_{jt}^{(5)})) \right) \right],$$

and we have defined

$$\psi_{jt}^{(1)} = \boldsymbol{w}_{1j}^{\top} \boldsymbol{z}_{t-1} + \boldsymbol{w}_{3j}^{\top} \boldsymbol{h}_{t-1} + b_{1j}, \tag{27}$$

$$\psi_{kt}^{(2)} = \boldsymbol{w}_{2k}^{\top} \boldsymbol{z}_t + \boldsymbol{w}_{4k}^{\top} \boldsymbol{h}_{t-1} + \boldsymbol{w}_{6k}^{\top} \boldsymbol{v}_{t-1} + b_{2k}, \tag{28}$$

$$\psi_{mt}^{(3)} = \boldsymbol{w}_{5m}^{\top} \boldsymbol{h}_t + \boldsymbol{w}_{7m}^{\top} \boldsymbol{v}_{t-1} + b_{3m}, \tag{29}$$

$$\psi_{kt}^{(4)} = \boldsymbol{u}_{5k}^{\top} \boldsymbol{v}_t + \boldsymbol{u}_{4k}^{\top} \boldsymbol{h}_{t-1} + \boldsymbol{u}_{6k}^{\top} \boldsymbol{v}_{t-1} + c_{2k}, \tag{30}$$

$$\psi_{jt}^{(3)} = \boldsymbol{u}_{2j}^{\top} \boldsymbol{h}_t + \boldsymbol{u}_{1j}^{\top} \boldsymbol{z}_{t-1} + \boldsymbol{u}_{3j}^{\top} \boldsymbol{h}_{t-1} + c_{1j}. \tag{31}$$

All the gradients can be calculated readily as in (9)-(12).

## C.2 Using deterministic hidden layer

For the generative model, denote the deterministic hidden layer as $\boldsymbol{h}_t^g \in \mathbb{R}^K$. For the recognition model, denote the deterministic hidden layer as $\boldsymbol{h}_t^r \in \mathbb{R}^K$. $\mathbf{W}_3$ and $\mathbf{U}_3$ are set to be zero matrices for the ease of gradient calculation. The generative model is expressed as

$$p(z_{jt} = 1) = \sigma(\boldsymbol{w}_{1j}^{\top} \boldsymbol{z}_{t-1} + b_{1j}), \tag{32}$$

$$h_{kt}^g = f(\boldsymbol{w}_{2k}^{\top} \boldsymbol{z}_t + \boldsymbol{w}_{4k}^{\top} \boldsymbol{h}_{t-1}^g + \boldsymbol{w}_{6k}^{\top} \boldsymbol{v}_{t-1} + b_{2k}), \tag{33}$$

$$p(v_{mt} = 1) = \sigma(\boldsymbol{w}_{5m}^{\top} \boldsymbol{h}_t + \boldsymbol{w}_{7m}^{\top} \boldsymbol{v}_{t-1} + b_{3m}), \tag{34}$$

The corresponding recognition model is expressed as

$$h_{kt}^r = f(\boldsymbol{u}_{5k}^{\top} \boldsymbol{v}_t + \boldsymbol{u}_{4k}^{\top} \boldsymbol{h}_{t-1} + \boldsymbol{u}_{6k}^{\top} \boldsymbol{v}_{t-1} + c_{2k}) \tag{35}$$

$$q(\boldsymbol{z}_{jt} = 1) = \sigma(\boldsymbol{u}_{2j}^{\top} \boldsymbol{h}_t + \boldsymbol{u}_{1j}^{\top} \boldsymbol{z}_{t-1} + c_{1j}) \tag{36}$$

Hence, $\mathcal{L} = \sum_{t=1}^{T} \mathbb{E}_{q_\phi(\boldsymbol{h}|\boldsymbol{v})}[l_t]$, and $l_t$ is expressed as

$$l_t = \sum_{j=1}^{J} \left( \psi_{jt}^{(1)} z_{jt} - \log(1 + \exp(\psi_{jt}^{(1)})) \right) + \sum_{m=1}^{M} \left( \psi_{mt}^{(2)} v_{mt} - \log(1 + \exp(\psi_{mt}^{(2)})) \right) \tag{37}$$

$$- \left[ \sum_{j=1}^{J} \left( \psi_{jt}^{(3)} z_{jt} - \log(1 + \exp(\psi_{jt}^{(3)})) \right) \right],$$

and we have defined

$$\psi_{jt}^{(1)} = \boldsymbol{w}_{1j}^{\top} \boldsymbol{z}_{t-1} + b_{1j}, \tag{38}$$

$$\psi_{mt}^{(2)} = \boldsymbol{w}_{5m}^{\top} \boldsymbol{h}_t^g + \boldsymbol{w}_{7m}^{\top} \boldsymbol{v}_{t-1} + b_{3m}, \tag{39}$$

$$\psi_{jt}^{(3)} = \boldsymbol{u}_{2j}^{\top} \boldsymbol{h}_t^r + \boldsymbol{u}_{1j}^{\top} \boldsymbol{z}_{t-1} + c_{1j}. \tag{40}$$

The gradients *w.r.t.* $\mathbf{W}_1, \mathbf{W}_5, \mathbf{W}_7, \mathbf{U}_1$ and $\mathbf{U}_2$ can be calculated easily. In order to calculate the gradients *w.r.t.* $\mathbf{W}_2, \mathbf{W}_4, \mathbf{W}_6, \mathbf{U}_4, \mathbf{U}_5$ and $\mathbf{U}_6$, we need to obtain $\frac{\partial \mathcal{L}}{\partial h_{kt}^g}$ and $\frac{\partial \mathcal{L}}{\partial h_{kt}^r}$, which can be

Table 1: Average prediction error and the average negative log-likelihood per frame for the bouncing balls dataset. ($\diamond$) taken from [2].

| MODEL | DIM | ORDER | PRED. ERR. | NEG. LOG. LIKE. |
|---|---|---|---|---|
| DTSBN-S | 100-100 | 2 | **2.79** $\pm$ 0.39 | **69.29** $\pm$ 1.52 |
| DTSBN-D | 100-100 | 2 | 2.99 $\pm$ 0.42 | 70.47 $\pm$ 1.52 |
| DTSBN-S | 100-100 | 1 | 10.39 $\pm$ 0.38 | 78.63 $\pm$ 0.92 |
| TSBN | 100 | 4 | 3.07 $\pm$ 0.40 | 70.41 $\pm$ 1.55 |
| TSBN | 100 | 2 | 4.00 $\pm$ 0.45 | 73.32 $\pm$ 1.75 |
| TSBN | 100 | 1 | 9.48 $\pm$ 0.38 | 77.71 $\pm$ 0.83 |
| HMSBN | 100 | 1 | 23.94 $\pm$ 0.41 | 86.27 $\pm$ 0.80 |
| AR | 0 | 2 | 3.63 $\pm$ 0.42 | 73.80 $\pm$ 1.46 |
| AR | 0 | 1 | 11.01 $\pm$ 0.24 | 93.61 $\pm$ 0.67 |
| RTRBM$^{\diamond}$ | 3750 | 1 | 3.88 $\pm$ 0.33 | $-$ |
| SRTRBM$^{\diamond}$ | 3750 | 1 | 3.31 $\pm$ 0.33 | $-$ |

calculated recursively via the back-propagation through time algorithm. Specifically, $\frac{\partial \mathcal{L}}{\partial h_{kt}^g} = \frac{\partial \mathcal{Q}_1}{\partial h_{kt}^g}$ and we have defined

$$\mathcal{Q}_1 = \sum_{t=1}^{T} \sum_{m=1}^{M} \left( \psi_{mt}^{(2)} v_{mt} - \log(1 + \exp(\psi_{mt}^{(2)})) \right) . \tag{41}$$

We observe that $\mathcal{Q}_1$ can be computed recursively using

$$\mathcal{Q}_t = \sum_{\tau=t}^{T} \sum_{m=1}^{M} \left( \psi_{m\tau}^{(2)} v_{m\tau} - \log(1 + \exp(\psi_{m\tau}^{(2)})) \right) \tag{42}$$

$$= \mathcal{Q}_{t+1} + \sum_{m=1}^{M} \left( \psi_{mt}^{(2)} v_{mt} - \log(1 + \exp(\psi_{mt}^{(2)})) \right) , \tag{43}$$

where $\mathcal{Q}_{T+1} = 0$. Using the chain rule, we have

$$\frac{\partial \mathcal{Q}_t}{\partial h_{kt}^g} = \sum_{k'} \frac{\partial \mathcal{Q}_{t+1}}{\partial h_{k't+1}^g} \cdot \frac{\partial h_{k't+1}^g}{\partial h_{kt}^g} + \sum_{m=1}^{M} w_{5mk}(v_{mt} - \sigma(\psi_{mt}^{(2)})) \tag{44}$$

$$= \sum_{k'} \frac{\partial \mathcal{Q}_{t+1}}{\partial h_{k't+1}^g} \cdot f'(\psi_{k't+1}^{(4)}) w_{4k'k} + \sum_{m=1}^{M} w_{5mk}(v_{mt} - \sigma(\psi_{mt}^{(2)})) , \tag{45}$$

where we have defined

$$\psi_{kt}^{(4)} = \boldsymbol{w}_{2k}^{\top} \boldsymbol{z}_t + \boldsymbol{w}_{4k}^{\top} \boldsymbol{h}_{t-1}^g + \boldsymbol{w}_{6k}^{\top} \boldsymbol{v}_{t-1}^g + b_{2k} , \tag{46}$$

and

$$\frac{\partial \mathcal{Q}_T}{\partial h_{kt}^g} = \sum_{m=1}^{M} w_{5mk}(v_{mT} - \sigma(\psi_{mT}^{(2)})) . \tag{47}$$

$\frac{\partial \mathcal{L}}{\partial h_{kt}^r}$ can be calculated similarly.

# D  Additional Results

## D.1  Generated Data

The generated, synthetic motion capture data, and polyphonic music data can be downloaded from `https://drive.google.com/drive/u/0/folders/0B1HR6m3IZSO_SWt0aSloYmlneDQ`.

## D.2  Bouncing balls dataset

Additional experimental results are shown in Table 1. AR represents an auto-regressive Markov model without latent variables [3].

Table 2: Average prediction error obtained for the MIT motion capture dataset.

| MODEL | PRED. ERR. |
|---|---|
| DTSBN-S | **3.71** $\pm$ 0.03 |
| DTSBN-D | 4.19 $\pm$ 0.01 |
| TSBN | 3.86 $\pm$ 0.02 |
| HMSBN | 17.49 $\pm$ 0.20 |

## D.3  MIT motion capture dataset

We randomly select 10% of the dataset as the test set. Quantitative results are shown in Table 2.