[Reviews · NeurIPS 2015]

Submitted by Assigned_Reviewer_1

This paper proposes a model for time series based on a hierarchy of sigmoid belief networks connected through time. The recently proposed Neural Variational Inference and Learning (NVIL) framework is applied to design scalable (approximate) inference and learning in the model. The model is shown to generate bouncing balls, polyphonic music, motion capture and text.

Strong points - The starting point of this paper (deep directed models, variational inference) is a popular and interesting line of research. This is one paper amid several time series extensions of this family of models/inference methods and is a natural extension of this line of work. This model takes a different approach to the time series extension than the other papers I have read to-date. - The paper is generally clear and provides more details in supplementary material - The paper considers most of the popular datasets/data modalities for "deep learning" time series and shows the model working reasonably well (perhaps, except for the polyphonic music) compared to other distributed-state time series models

Weak points - The paper fails to cite other recent papers in this area, mainly recurrent extensions of variational autoencoders - The approach is valid and interesting but not really that creative as it takes a known directed model (sigmoid belief net), connects it through time and applies a known inference/learning approach (NVIL)

Specific comments - "RBM... parameter estimation relies on an approximation technique known as contrastive divergence" -- I would insert "usually" here before relies; there are many alternatives to CD, though not as popular - "...our formalism uses a diagonal matrix to parameterize the covariance structure..." if it's diagonal wouldn't there be no covariance structure and this would simply be variance structure? - In the related work section, please include some recent temporal extensions of SGVB-type models, e.g.

* Bayer and Osendorfer, Variational inference of latent state sequences using Recurrent Networks

* Fabius and van Amersfoot, Variational Recurrent Auto-Encoders

* Chung et al., A Recurrent Latent Variable Model for Sequential Data - Best results are bolded in Table 2 and 4, but not table 3

Summary: A straightforward and apparently effective extension of Sigmoid Belief Networks to time series modeling which utilizes the Neural Variational Inference and Learning approach.

Submitted by Assigned_Reviewer_2

The paper trains a recurrent network with binary latent variables. The NVIL algorithm is used to estimate the gradients.

This can be a great paper, if the experiments are more systematic. 1) The average prediction error is not a good measure when modeling a multi-modal distribution. The log-likelihood or the variational lower bound should be used instead. I understand that the RBM results are without log-likelihood. You can still use log-likelihood to compare your architectures.

2) It is not clear how much are the latent variables helping. A baseline should be a model with a high order and no latent variables.

3) It is not clear whether the depth is helping or if similar results can be obtained by using more hidden units.

Hints for architecture improvements: 4) The P(h_t|h_{t-1}, v_{t-1}) conditional probability distribution can be a deeper non-linear function.

5) The description of the model with the deterministic hidden layer is not clear. Was a recurrent neural network used or were only skip connections used?

Related work: 6) A similar architecture was proposed in "The Helmholtz Machine Through Time" by Geoff Hinton etc.

Future: 7) It would be interesting to compare your model on the datasets from "A Recurrent Latent Variable Model for Sequential Data" by Junyoung Chung etc.
Summary: The paper has a chance to establish a baseline for modeling of temporal sequences. The different model architectures would deserve to be compared based on log-likelihood.

Submitted by Assigned_Reviewer_3

I enjoyed reading the paper. It presents some interesting applications of stochastic variational inference which distinguish it from the usual paper. Methodologically, the paper defines a recurrent sigmoid belief network to model sequential data. Since maximum likelihood learning of the model is difficult, the authors propose to use a variational setup, where the approximating distribution q is defined as another directed model. A lower bound on the log likelihood of data is maximized by alternatively maximizing the joint distribution p(v,h) under samples h from q, and maximizing parameters of q in equation 11 (which I believe is equivalent to minimizing the KL between posterior of p and q, although its never stated that way). The optimization is done with Stochastic Gradient Descent on Monte Carlo samples (as was popularized in Stochastic Variational Inference by Hoffman, Blei et. al and other papers predating NVIL) since exact analytical treatment is not possible.

The results are presented on four different datasets with interesting results.

With regards to the results, I was surprised that the TSBN performed quite poorly on bouncing balls dataset with an order of 1 (pred err 13.23), but so well with order of 4 (pred err = 3.07). To me this implies that the model has not done a very good job of learning long range structure, through the hidden units. No results are presented for the deep TSBM with order 1. And yet these are compared to the temporal restricted boltzmann machine, which I believe used order 1. Either ways, it would seem that RTRBM actually performs better on bouncing balls than this model if order 1 is used.

For Motion Capture results it would have been good to also report results from conditional RBM models (although, I believe the MSE results are worse than those in table 2). The conditional RBMs models videos seem a lot less jittery than the ones attached in supplementary methods for this paper.

The presentation of section 5.4 [Prediction] is a little incomplete. It isn't stated what the modeling task is clearly, or what the data is that is modeled at each time step. Presumably its the word distribution at each time step, and this information is available in the supplied reference [31]. However it would be good to have at least a one line description here, space permitting. Similarly for the Dynamic Topic Modeling subsection.

Quality: Paper is well presented, and methods are demonstrated on datasets with quite different characteristics, showing that the models is versatile enough. Clarity: Quite clear presentation of ideas. I would suggest one thing. Wrt equation 7, authors should state that this procedure optimizes the exact bound if q is exactly the posterior of p. Also, it should be stated that optimizing equation 11 is equivalent to minimizing the KL of posterior of p from q. For clarity, a one line note with equation 11 can be added to state that sum_h q(h) d/dtheta log q(h) is 0 and drops out, in equation 11.

Originality: The novelty of the paper is in defining the temporal SBM and pairing it with the recognition network. The other novelty is in the application to the different datasets. The methods themselves aren't entirely new, but are an application of neural stochastic variational inference to this model. However, the paper is novel enough, in my opinion, since it shows this method can learn.

Significance: The paper extends the range of successful applications to which stochastic variational inference ideas have been applied, and is hence useful.

Nevertheless, it would have been useful to compare with other variational idea that do sequence learning (Learning Stochastic Recurrent Networks by Bayer and Oserndorfer).

Summary: The paper presents a relatively straightforward application of stochastic variational inference ideas, that are currently in vogue, to learning temporal sigmoid belief networks. The paper is quite well presented and the methods are demonstrated on four different datasets with different conditional distributions of data. Overall, a decent paper, with reasonable results. My recommendation would be higher save for the fact that some of the results were not as nice as I would have hoped. For example the videos from motion capture shows a high frequency jitter in movements, and bouncing ball results seem quite poor when order 1 recurrence is used - results that worse than an order 1 RTRBM.

Submitted by Assigned_Reviewer_4

This paper proposes a learning and inference method for the deep high order temporal sigmoid belief networks, which can be seen as an extension of the neural variational inference and learning method [13] to sequential data. The distribution of the current time step is conditioned on the states in previous several time steps.

Parameter learning is done by maximizing the lower bound of the data log-likelihood through an inference network.

Advantages:

1. The extension from deep sigmoid belief network to high order temporal SBN is novel.

2. The authors also extend their model to both multi-value discrete data and continuous data.

Disadvantages:

1. The learning and inference method is quite similar to that of NVIL [13] in terms of three aspects:

(1) Both use the same objective function, i.e., the lower bound of the marginal log likelihood, and the same inference network to approximate the inference during learning.

(2) The ideas of drawing samples from the inference network to estimate the gradients are the same.

(3) The variation reduction techniques are the same.

2. It is unclear from the paper how the model is learnt. Does a sliding window go through the sequence to obtain n frames if the order of the model is n-1? And then the actual learning is performed for a larger size SBN?

3. As a dynamic Bayesian Network, temporal SBN should consist of a prior model and a transition model. They have different parameters and are learnt from different data. The authors so far only learn the transition model and the prior model is not learnt. What is the justification for ignoring the prior model?

4. About experiments, for the first experiment on tracking, the authors should also compare with the shallow LDS model, besides deep dynamic models. For the second experiment on generating new sequence on MIT dataset, the experiment is qualitative consisting of only visual observation, without any quantitative comparison. Also, how is the model initialized? Is the initial frames sampled from the model or manually set? What's more, a common way to evaluate a dynamic model is sequence classification. Since the likelihood of the model can be efficiently estimated, can the model proposed in this paper be used for classification if one model is trained for one class?

============================

In the authors' response, they explained how to define the prior model, provided more details about the learning process, and they promised to include more experiments. Although I still feel the experiment evaluation is not strong enough, some other concerns are addressed, therefore I upgrade my rating to 6.

Summary: This paper proposes a learning and inference method for the

temporal sigmoid belief networks by extending the NVIL method [13].

The algorithm is built upon an existing framework with incremental theoretical novelty.

Experiments are weak and unclear to demonstrate the benefits of the model for modeling dynamics.

Author Feedback
Author rebuttal: We thank the reviewers for their valuable and encouraging comments, and address their major questions and comments below. SM is short for supplementary material.

Reviewer 2:

On the bouncing balls experiment: We agree that the TSBN with order-1 performed poorly on this dataset, and the RTRBM seems to perform better. This is partially because only 100 hidden units were used in the TSBN, while 3750 hidden units were utilized in the RTRBM. Using a higher-order, or a deeper TSBN improved the performance. We have these additional results, including the deep TSBN with order-1, and will provide them in the SM.

On the mocap video generation: The synthetic motion videos in the SM were generated by an order-1 HMSBN model, whose prediction error is large in Table 2. This explains why some of the generated videos are jittery. We also generated data by using a deeper TSBN. The generated data are smoother, which will be added to the SM.

On the Clarification of Section 5.4 and Equation 7 & 11: We will make the description clearer as suggested. We will also cite the paper by Bayer and Oserndorfer.

Reviewer 3:

On the NVIL inference method: Our NVIL algorithm is the same as the one used in [13]. The focus in this work is on sequence modeling, and our approach serves as a new application of this algorithm for a SBN-based time-series model. The improvement of the NVIL algorithm itself can be a useful direction for future work.

On the learning details of the model: A sliding window was used to go through the sequence to obtain n frames. The first (n-1) frames are used to adjust the hidden bias in the SBN model of the nth frame. Learning is then performed on the same size SBN as what we have defined, not a larger one. This makes us different from the SBN model used in [13]. Thanks for the insightful suggestion. We will make the learning for the temporal model clearer.

On the prior model of the TSBN: The prior model p(h1), and its approximate posterior q(h1|data) is implicitly defined in Eq. 3 and 9, by setting h0 and v0 to zero vectors, and replacing the bias term b/d with another b0/d0. We will clarify this in the paper.

On the comparison with the LDS: We focus on comparing our deep dynamic models, with RBM-related models, since the SBN is closely related to the RBM (see Section 2.1). It would be interesting to also compare with the LDS, and we will add this LDS experiment as requested.

On the motion capture experiment: Quantitative results have been shown on the CMU dataset to demonstrate the superior prediction performance of our TSBN model (see Table 2). The prediction error results are not shown for the MIT dataset due to space limits, and will be provided in the SM. The model is initialized via randomly sampling, 2-5 frames from the training dataset, not manually set, this will be emphasized.

On the sequence classification: The focus of this paper is to study the prediction and generative ability of TSBN for an individual sequence. The study for the classification of multiple sequences when label information is given can be a very interesting future work to explore. Our model has the potential for classification.

On the benefits of the model for modeling dynamics: We have considered 3 popular benchmarks for sequence modeling in the deep learning literature, i.e. bouncing balls, motion caption datasets [7,8,9,10,11], and music dataset [10]. In all these papers, typically average prediction errors are calculated and the "fantasy" data are generated to show the benefits of the model for modeling dynamics. Beyond that, we provide an additional experiment on text streaming to show the versatility of the proposed model.

Reviewer 4:

We will cite all the recent recurrent extensions of variational autoencoders. We will also add "usually" before "relies", and change "covariance structure" to "variance structure", as suggested.

Reviewer 5:

On the performance evaluation: We agree that the log-likelihood is a better measure. We have log-likelihood results for all the datasets. The results roughly agree with what we have observed using the prediction error metric, and will be provided in the SM.

On the usage of latent variables: We will add the result of a model with high order and no latent variables. We believe introducing latent variables will help improve performance, but more importantly, help us understand the latent pattern underlying the data, e.g. the dictionaries shown in Figure 1(a).

On the model depth: Adding one more layer can generally improve the performance slightly (see Table 1 and 2). Comparing a model with more hidden units will also be interesting.

On the deterministic hidden layer: The top hidden layer is stochastic. The middle deterministic hidden layer is equivalent to a recurrent neural network. This combination of stochastic and deterministic layers is also considered in [14].

We will cite the paper by G. Hinton and the work by J. Chung.